# Baseline Imaging Derived Predictive Factors of Response Following [^177^Lu]Lu-PSMA-617 Therapy in Salvage Metastatic Castration-Resistant Prostate Cancer: A Lesion- and Patient-Based Analysis

**DOI:** 10.3390/biomedicines10071575

**Published:** 2022-07-01

**Authors:** Esmée C. A. van der Sar, Adinda J. S. Kühr, Sander C. Ebbers, Andrew M. Henderson, Bart de Keizer, Marnix G. E. H. Lam, Arthur J. A. T. Braat

**Affiliations:** 1Department of Radiology and Nuclear Medicine, University Medical Center Utrecht, 3584 CX Utrecht, The Netherlands; a.j.s.kuhr@student.vu.nl (A.J.S.K.); s.c.ebbers-2@umcutrecht.nl (S.C.E.); b.dekeizer@umcutrecht.nl (B.d.K.); m.lam@umcutrecht.nl (M.G.E.H.L.); a.j.a.t.braat@umcutrecht.nl (A.J.A.T.B.); 2Mercy Radiology, Auckland Central, Auckland 1010, New Zealand; ahenderson@radiology.co.nz

**Keywords:** prostate cancer, lutetium, radio-ligand therapy, prostate specific membrane antigen, predictors

## Abstract

Earlier studies have mostly identified pre-therapeutic clinical and laboratory parameters for the prediction of treatment response to [^177^Lu]Lu-PSMA-617 in metastatic castration resistant prostate cancer patients (mCRPC). The current study investigated whether imaging-derived factors on baseline [^68^Ga]Ga-PSMA-11 PET/CT can potentially predict the response after two cycles of [^177^Lu]Lu-PSMA-617 treatment, in a lesion- and patient-based analysis in men with mCRPC. Included patients had histologically proven mCRPC and a [^68^Ga]Ga-PSMA-11 PET/CT before and after two cycles of [^177^Lu]Lu-PSMA-617 treatment. The imaging-based response was evaluated on lesion-level (standardized uptake value (SUV) reduction) and patient-level (total lesion PSMA (TL-PSMA) reduction). In the lesion-level analysis, a clear relationship was found between SUV_peak/max_ and the imaging-based response to [^68^Ga]Ga-PSMA-11 PET/CT (most avid lesion SUV_peak/max_ ≥ 30% reduction) (*p* < 0.001), with no significant difference in cut-off values between different sites of metastases (i.e., lymph node, bone or visceral metastasis). In patient-level analysis, baseline PSA and SUV_peak_ values of most avid metastasis were significantly associated with imaging-based response (TL-PSMA ≥ 30% reduction) (*p* = 0.019 and *p* = 0.015). In pre-treatment with [^68^Ga]Ga-PSMA-11 PET/CT, a clear accumulation-response relationship in lesion-level was found for SUV_peak/max_ in men with mCRPC receiving two cycles of [^177^Lu]Lu-PSMA-617 treatment. The SUV_peak_ of the most avid lesion was the only image-derived factor predictive of the imaging-based response at the patient-level.

## 1. Introduction

Worldwide, prostate cancer was the third most common diagnosed malignancy in 2020 [1]. The survival rates of prostate cancer are subject to the degree of metastasis. The five-year survival of localized prostate cancer is 100%, however, it falls rapidly to 31% in patients with distant metastases [2].

In the last decade, new treatment options for patients with metastatic, castration resistant prostate cancer (mCRPC) became available, including novel androgen axis drugs (e.g., abiraterone, enzalutamide) and chemotherapy (i.e., docetaxel and cabazitaxel). More recently radioligand therapy with lutetium-177 prostate specific membrane antigen ([^177^Lu]Lu-PSMA-617) emerged as a promising treatment for advanced prostate cancer [3]. Several studies have demonstrated the safety and efficacy (extended overall survival and improved quality of life) of [^177^Lu]Lu-PSMA-617 treatment in mCRPC patients [3,4,5]. Although the majority (68%-75%) of the metastatic prostate cancer patients receiving [^177^Lu]Lu-PSMA-617 treatment showed some degree of response, some did not benefit [6].

Earlier studies mostly identified pre-therapeutic clinical and laboratory parameters for the prediction of treatment response (defined as prostate specific antigen (PSA) change) [7,8,9]. Only a sparse literature exists on imaging-derived predictors (defined by pre-treatment with [^68^Ga]Ga-PSMA-11 PET/CT), which has showed that low SUV_max_ and SUV_mean_ values were negative predictors for treatment response [10,11]. Unfortunately, these studies only evaluated treatment response on the patient-level by evaluating biochemical changes (PSA). The use of PSA reduction is the most commonly used in clinic due to its simplicity, however for response evaluation PSA is still under debate and cannot be used as response criterion in a per-lesion analysis [12]. This study investigated whether imaging-derived factors on baseline [^68^Ga]Ga-PSMA-11 PET/CT can potentially predict the response to [^177^Lu]Lu-PSMA-617 treatment, in a lesion- and patient-level analysis, in men with mCRPC.

## 2. Materials and Methods

### 2.1. Population

Patients referred for and treated with [^177^Lu]Lu-PSMA-617 were identified retrospectively from a single center from March 2017 to November 2019. Patients were included if they had histologically proven mCRPC and had a [^68^Ga]Ga-PSMA-11 PET/CT before and after two cycles of [^177^Lu]Lu-PSMA-617 treatment. The reason why we choose to analyze after two cycles is based on the findings of Ahmadzadehfar et al., who showed that response (PSA decline ≥ 50%) is only or mostly seen after the second cycle of [^177^Lu]Lu-PSMA-617 treatment [13].

Patients were excluded if the interval between the baseline and post treatment [^68^Ga]Ga-PSMA-11 PET/CT was more than eight months. Blood testing was performed at the time of admission. The PSMA-617 ligand was obtained from ABX GmbH, Radeberg, Germany. A total of 6.0 or 7.4 GBq [^177^Lu]Lu-PSMA-617 per 40 to 250 μg peptide was administered intravenously for each cycle, with a planned interval of six weeks.

The need for informed consent was waived by the institutions medical ethics committee for this retrospective study.

### 2.2. Image Acquisition and Reconstruction

Sixty minutes after intravenous administration of 1.5–2.0 MBq/kg [^68^Ga]Ga-PSMA-11 the imaging was performed from the skull vertex to the mid-thigh (Biograph mCT scanner, Siemens, Erlangen, Germany).

The PET reconstruction was obtained following the EANM Research Ltd. (EARL), Vienna, Austria recommendations although its use for [^68^Ga]Ga-PSMA-11 PET/CT interpretation has not been validated yet [14,15]. The [^68^Ga]Ga-PSMA-11 accumulation was corrected for lean body mass (in SUV_lbm,peak_*cm^3^) [16].

### 2.3. Imaging Analysis

Syngo.via-software (Siemens version 05.01, Erlangen, Germany) was used to establish quantitative image analysis. Based on PERCIST, relevant volumes of interest were (semi)automatically segmented if the standardized uptake value of peak (SUV_peak_) was greater than the threshold set by a 3 cm cylindrical volume of interest (VOI) in the aorta with a threshold of 1.5 × aorta peak + 2 × standard deviation [17]. The activity in the blood pool has been shown to be a well-grounded reference region for [^68^Ga]Ga-PSMA-11 imaging interpretation [18].

Manual adoption was needed if single tumor lesions and organs were not automatically divided based on the set PERCIST criteria.

Segmentations smaller than 0.3 mL where disregarded.

The Syngo.via software only allowed visual validation of a maximum of 50 lesions on the [^68^Ga]Ga-PSMA-11 PET/CT. It automatically calculated the total amount of [^68^Ga]Ga-PSMA-11 accumulation of the remaining lesions (>50).

Parameters collected included PSMA tumor volume (PSMA-TV) in mL and TL-PSMA (summation of the entire tumor load within the patient derived from total lesion glycolysis (TLG)). The SUV_peak_ and SUV_max_ of the primary prostate tumor (if in situ), and the SUV_peak_ and SUV_max_ the two most- and least-avid lesions of three different organ categories (lymph nodes, bone and visceral metastasis) were collected. This approach was chosen in order to collect a wide variety of lesion avidity for the lesion-based analysis.

In accordance with the EARL recommendations, the TL-PSMA was calculated by multiplying the SUV_peak_ value with the PSMA-TV (SUV_lbm,peak_*cm^3^) per patient. Although the EARL recommendations are used for 18F-FDG (FDG), the used method for the total lesion glycolysis (TLG) will best represent the in vivo distribution of [^68^Ga]Ga-PSMA-11 by calculating the fractional tumor activity [19,20,21].

### 2.4. Outcomes

The primary outcome of this study was defined as an objective response after two cycles of [^177^Lu]Lu-PSMA-617 treatment at the lesion- and patient-level. Response evaluation at the lesion-level was based on PERCIST [17]: imaging complete response (iCR); complete resolution of PSMA-tracer accumulation in all lesions, imaging partial response (iPR); more than or equal to 30% reduction of SUV_peak_, imaging progressive disease (iPD); more than or equal to 30% increase in SUV_peak_ and imaging stable disease (iSD); not qualifying for iCR, iPR, or iPD. The definition of objective response includes iCR + iPR. For the patient-based analysis, the same methodology was used, except the TL-PSMA was used as the distinctive parameter instead of the SUV_peak_, thus objective response at the patient-level was determined as a reduction of TL-PSMA ≥30% and progressive disease was defined as more than or equal to 30% increase in the TL-PSMA and/or the appearance of new lesions.

Secondary outcomes included a biochemical response after two cycles of [^177^Lu]Lu-PSMA-617 treatment at the patient-level, defined according to the prostate cancer clinical trial working group 2 and 3 [22,23]. Response definitions: a partial response (bPR) was more than 50% PSA level reduction; progressive disease (bPD) was more than or equal to 25% increase; and a stable disease (bSD) was less or equal to 50% reduction and less than 25% increase of PSA level.

Additionally, clinical, biochemical, imaging, and hematological parameters (Table 1 and Table 2, and Appendix B Table A1) were gathered to investigate potential predictive factors on a patient-level.

Finally, overall survival (OS), defined as after the first cycle of [^177^Lu]Lu-PSMA-617 treatment to death from any cause, was analyzed on a patient-level.

### 2.5. Statistical Analysis

The software IBM SPSS Statistics version 25.0.0.2 for Windows (IBM, Armonk, NY, USA) and R version 4.0.1 (R Core Team 2020) was used for all analyses (for used R codes see Appendix A). As accumulation measurements (i.e., SUV_peak_ and SUV_max_) showed positive skewness, a log-transformation was applied before analyses were executed. Several types of analyses were executed to test different hypotheses. A *p*-value ≤ 0.05 was considered significant.

For our primary outcome, the imaging-based response on a lesion-level, Receiver Operating Characteristics (ROC) curve analyses were performed for predicting imaging-based response, including the following variables: SUV_peak_/SUV_max_ all measured lesions together, SUV_peak_/SUV_max_ lymph node metastases and bone metastases. The Youden’s index test and a set minimum specificity of 0.80 were used to determine the optimal cut-off value for binarization of the predictive values.

Mixed-effects models with SUV_peak_/SUV_max_ as the independent variable, imaging-based response (dichotomized or as categorical variable) as the dependent variable, and a random intercept of SUV_peak_ and SUV_max_ per patient was used to model the effect of imaging-based response on the SUV_peak_ and SUV_max_ values. The random intercept was added to the model to incorporate the anticipated between-patient variation in SUV_peak_ and SUV_max_ levels. The remaining dependent relation between the imaging-based response and SUV_peak_ or SUV_max_ was then modelled as a fixed effect. PSMA-TV in the patient and metastasis type were added to the model as a confounder. In order to test the hypothesis that type of metastasis (i.e., lymph node, bone and visceral metastases) was of influence on the relationship between the individual pre-treatment accumulation and imaging-based response per lesion, metastasis type was added as an interaction to the model, both as a categorical variable and as a dichotomized variable in separate models.

For the imaging-based response on a patient-level, the maximum SUV_peak_ and SUV_max_ values in primary tumor or metastases (lymph node, bone, and visceral metastases) were tested for a relationship with response in logistic regression analysis. This approach was also used for the secondary outcome, biochemical response.

Several variables were tested in logistic regression analysis, univariately, while correcting for tumor load by including baseline PSMA-TV in each model. Each model was tested using likelihood ratio tests, comparing the model including the variable with a model only including baseline PSMA-TV.

Overall survival analysis was done using Cox-proportional hazard models, and Kaplan–Meier survival curves were constructed. The ISUP Gleason score, ECOG performance score, extent of disease, and imaging parameters were included in Cox-proportional hazard regression.

## 3. Results

A total of 87 patients were treated with [^177^Lu]Lu-PSMA-617. Of those patients, 32 were eligible for analysis as illustrated in Figure 1.

The baseline patients and imaging parameters are summarized in Table 1 and Table 2. Baseline hematological parameters and radiopharmaceutical characteristics can be found in Appendix B, Table A1 and Table A2. A total of 86 lymph nodes, 119 bone, and 17 visceral metastases were extracted. Table 3 represent the imaging-based response rates on a patient- and lesion-level. Figure 2 illustrates the response rates of two patients after two cycle of [^177^Lu]Lu-PSMA-617 treatment (one responder and one non-responder).

### 3.1. Lesion-Level

In ROC analysis (Figure 3), the optimal cut-off value to predict imaging response based on Youden’s index was 14.87 for SUV_peak_ (sensitivity = 0.36, specificity = 0.90) and 19.08 for SUV_max_ (sensitivity = 0.40, specificity = 0.89). The cut-off values based on a minimum specificity of 0.80 were 12.07 (for SUV_peak_; sensitivity = 0.44) and 15.4 (for SUV_max_; sensitivity = 0.49), meaning that of all non-responding tumors, 80% showed accumulation below these cut-off values.

The relationship between baseline PET parameters and imaging-based response in linear mixed effects models were significant for both SUV_peak_ and SUV_max_, when testing for dichotomized imaging response (*p <* 0.001 and *p <* 0.001) and categorical imaging response (*p <* 0.001 and *p <* 0.001; Table 4). On average, in responding tumors a 1.80 (95% CI [1.42–2.29]) times higher SUV_peak_ and a 1.84 (95% CI [1.46–2.31]) times higher SUV_max_ was observed (Table 4 and Table 5).

The type of lesion (i.e., lymph node, bone, visceral metastasis, or primary prostate) did not alter the found relationships. Figure 4 and Figure 5 shows that lesions with a higher accumulation (SUV_peak_) at baseline have a better imaging-based response, with the exception of complete response.

### 3.2. Patient-Level

Results of the response evaluation on the patient-level are shown in Table 6. Baseline PSA (median 210.0 ng/mL) and SUV_peak_ most avid metastases had a significant relationship with imaging-based response (OR 2.07, *p* = 0.019 and OR 1.11, *p* = 0.015). Imaging-based response was highly associated with biochemical response (*p <* 0.001). Secondary, no factors were identified having a significant relationship with biochemical response.

During follow-up, 28 (87.5%) patients were found to have died with a median overall survival of ten months (Table 2). Log-rank testing showed that patients with an ECOG performance score of zero or one have a significant better survival rate than patients with an ECOG performance score of two (*p* = 0.033), the same for patients with >50% PSA reduction in comparison to no PSA reduction of >50% (*p* = 0.05) and for patients with ≥30% TL-PSMA reduction in comparison to no reduction of ≥30% (*p* = 0.048). Patients who had the primary tumor in situ had a better overall survival (*p* = 0.006) (Figure 6). The factors ISUP, presence of visceral metastases, bone metastases or lymph node metastases, baseline TL-PSMA, baseline PSMA-TV and most avid lesion SUV_peak_, SUV_max_ did not shown any significance.

In the univariate Cox-regression analyses, patients with an ECOG performance score of two and an unknown Gleason (ISUP) score had a significant hazard ratio (HR) (Table 7), but overall, these variables were not significantly associated with survival (*p* = 0.104 and *p* = 0.386). Biochemical response was significantly associated with survival (HR 0.43, *p* = 0.047; Table 7).

## 4. Discussion

This study evaluated the potential of imaging-derived factors on [^68^Ga]Ga-PSMA-11 PET/CT to predict the response in a lesion- and a patient-based analysis, in men with mCRPC receiving two cycles of [^177^Lu]Lu-PSMA-617 treatment. In the lesion-level analysis, a clear relationship was found between pre-therapeutic accumulation (SUV_peak_ and SUV_max_) and imaging-based response on [^68^Ga]Ga-PSMA-11 PET/CT with no preference or difference for either, primary tumor, lymph node, bone or visceral metastasis.

Interestingly, in the lesion-level analysis, a contradictory lower SUV_peak_ at baseline was seen in lesions with iCR. An explanation might be the threshold method based on PERCIST and lesion selection criteria in this study. If, at follow-up imaging, the lesion SUV_peak_ was under the threshold, it was set to zero (being iCR), even when visually some accumulation might still be present. An additional explanation can be found in the set threshold and making the accumulation of the least avid lesion depended on the blood pool activity. However, to address a large number of lesions with a wide variety of intensities, this approach was deliberately chosen to gain more insight in lesion-based response. A third explanation is the influence of partial volume effect on small lesions, potentially overestimating objective response.

In the clinical setting, a difference was noticed in the objective response between different lesion types (e.g., prostate, lymph node, bone or visceral), however, the results of this study show no difference between lesion type (Figure 4 and Figure 5). Thus, making a distinction between lesion types seems irrelevant for patient selection prior to ^177^Lu-PSMA-617. To the best knowledge of the authors, this is the first study evaluating the response on individual lesion-level with imaging-derived predictive factors on [^68^Ga]Ga-PSMA-11 PET/CT, thus no comparison with existing literature can be made.

On the patient-level, the pre-therapeutic imaging-derived predictive factor, SUV_peak_ of the most avid metastases was significantly associated with the imaging-based response (TL-PSMA ≥ 30% reduction). No other study has evaluated the imaging-based response based on PERCIST. Hofman et al. [5] and Sartor et al. [3], however, used the RECIST criteria, but in comparison to PERCIST (which looks at accumulation reduction); RECIST only uses single dimension size changes and has severe limitations in measuring bone or bone marrow disease [17]. On the other hand, evaluating tumor response with [^68^Ga]Ga-PSMA-11 PET/CT (via the same mechanism of the treatment itself) can also be debated, as potential non-PSMA avid disease will not be evaluated.

There are studies evaluating imaging-derived predictive factors with biochemical response as outcome, although the results are contradictory. Some did not find any significant imaging-derived predictive factor (e.g., SUV_max_ and SUV_mean_, total tumor load, number of metastatic lesions, and sites of disease) for biochemical response [11,24], while some did (e.g., SUV_max_ < 45 of the most avid lesion, SUV_mean_) [10,25]. An explanation can be found in the difference in the number of cycles used, population size, the used therapeutic radiopharmaceuticals, and the amount of activity, thereby making their results difficult to compare and to interpret.

In the survival analyses, patients with an ECOG performance score of zero and one had a significant better OS than patients with an ECOG performance score of two, in line with previous findings [7]. Furthermore, patients with a biochemical response (>50% PSA reduction) had a better OS, compared to biochemical non-responders. This is in contrast to the findings of Ahmadzadehfar et al. [7] and Rahbar et al. [9], who did not find a significant difference in 100 and 104 mCRPC patients treated with [^177^Lu]Lu-PSMA-617 between biochemical responders and non-responders concerning OS. The difference can possibly be explained by differences in sample size, population heterogeneity and selection bias by the range in the number of given [^177^Lu]Lu-PSMA-617 cycles in both studies: namely two to six in this study versus one to eight cycles.

This study has several limitations: first, the retrospective design and resulting missing data. Second, the small sample size (Figure 1) and population heterogeneity limits the ability to draw definite conclusions on patient-based analyses. But in the lesion-based analysis, the sample size was a total of 237 lesions, however, only the two most avid and two least avid lesions were selected to address many lesions with a wide variety of intensities, introducing a selection bias. Third, no volumes of the individual lesions were measured, introducing bias by partial volume effects. This could have influenced the response rate on the lesion-level analysis, as a lower PSMA-TV with the same SUV_peak_ is more prone for iCR than a higher PSMA-TV with the same SUV_peak_. On the patient-level, however, baseline TL-PSMA and PSMA-TV had no significant influence on the imaging-based response rate. Fourth, PERCIST for the lesion-based response evaluation is not validated for PSMA PET/CT. However, it is already broadly available in clinical practice and easy to apply [15,17].

Other studies used a maximum intensity threshold with SUV_max_ for tumor segmentation [20,26]. In this study, we chose to use SUV_peak_ adapted from PERCIST for tumor segmentation as this limits the influence of noise on quantification [27].

In current practice, patients are only eligible for [^177^Lu]Lu-PSMA-617 treatment if sufficient tracer accumulation is observed on PSMA PET/CT. However, the definition of sufficient tracer accumulation is still a topic of discussion. Currently, it is based on literature on peptide receptor radionuclide therapy (PRRT), as was also used in the VISION trial [28,29,30]: accumulation in tumor sites must at least be higher than physiological accumulation in normal liver tissue, to ensure a certain efficacy. The included patients in this study all met this specific criterion. Still, there were some non-responders (iPD and bPD) in this study (5/32; 16% and 5/30; 17%), in line with the findings in the VISION trial [3], thereby indicating that the decision whether or not an individual patient is eligible for [^177^Lu]Lu-PSMA-617 based on the visual assessment of accumulation alone compared to healthy liver tissue accumulation remains questionable. The results in this study indicate that tracer accumulation based on SUV_peak_ (>14.87) or SUV_max_ (>19.08) in a lesion can be helpful to determine if a certain lesion will or will not respond, based on a broadly available, internationally accredited image reconstruction method (EARL) [14]. In case, when all or the majority of metastases within a patient are below these thresholds, an alternative treatment may be more beneficial, subsequently, improving patient selection for ^177^Lu-PSMA-617 based on available pre-treatment [^68^Ga]Ga-PSMA-11 imaging.

The results of this study illustrate the potential of response prediction by pre-treatment [^68^Ga]Ga-PSMA-11 PET/CT quantification, using widely available image reconstruction parameters (EARL) and software packages enabling (semi-automated) PERCIST assessments. The findings of this study need to be validated in larger cohorts and future prospective studies.

## 5. Conclusions

On pre-treatment [^68^Ga]Ga-PSMA-11 PET/CT, a clear accumulation-response relationship in lesion-level analyses has been found for SUV_peak_ and SUV_max_ in men with mCRPC receiving two cycles of [^177^Lu]Lu-PSMA-617 treatment. On a patient-level analysis, SUV_peak_ of the most avid lesion was the only image-derived factor predictive of imaging-based response.

## Figures and Tables

**Figure 1 biomedicines-10-01575-f001:**
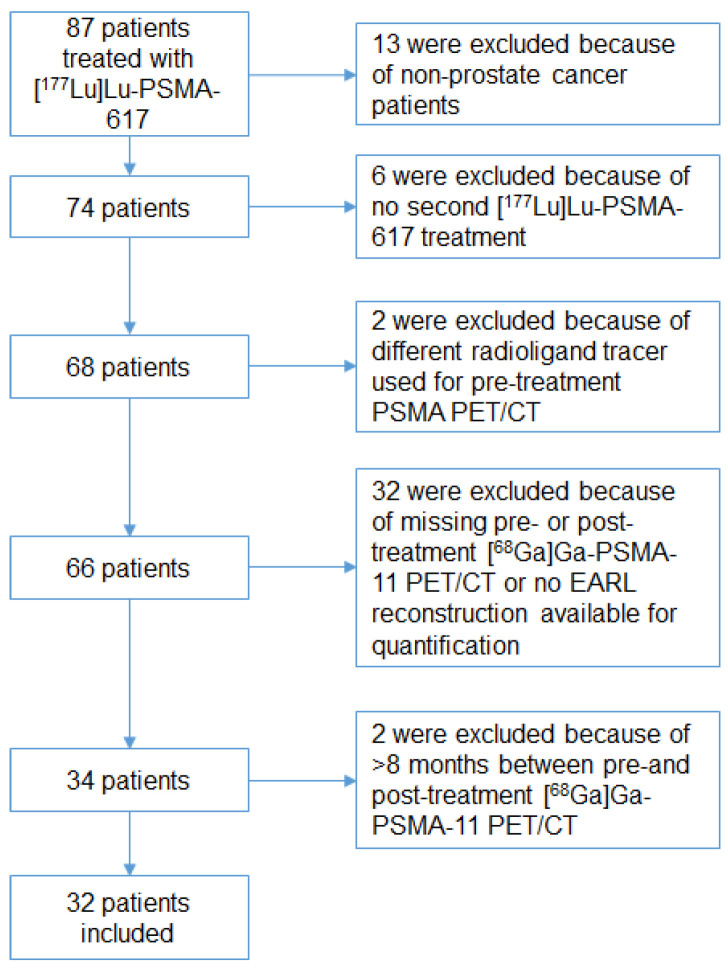
Flow-chart of the retrospective included patients.

**Figure 2 biomedicines-10-01575-f002:**
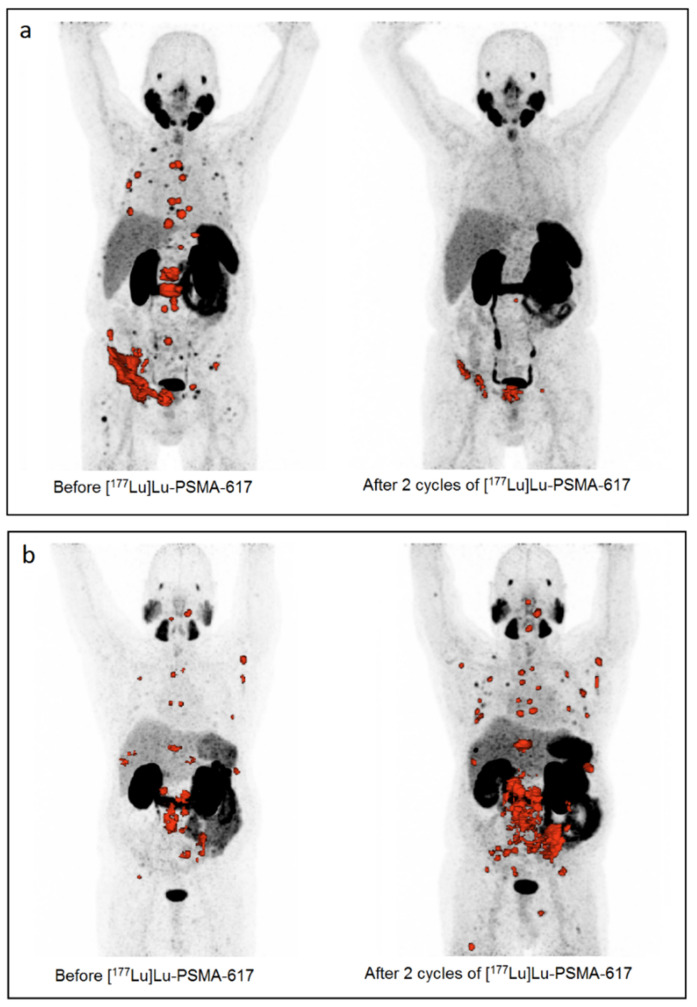
Example of two patients: responder and non-responder after two cycles of [^177^Lu]Lu-PSMA-617 treatment. (**a**): Responder (TL-PSMA reduction 95.01%, PSA-reduction: 99.5%): 78-year-old men with a Gleason score of nine, a ECOG performance score zero and a SUV_peak_ of the most avid lesion of 17.7. Activity first cycle [^177^Lu]Lu-PSMA-617: 6.3 GBq, activity second cycle [^177^Lu]Lu-PSMA-617: 6.3 GBq. TL-PSMA pre-treatment: 1961.02 SUV_lbm,peak_*cm^3^, TL-PSMA post-treatment: 97.79 SUV_lbm,peak_*cm^3^. (**b**): Non-responder (TL-PSMA increase: 750.77%, PSA-increase: 566.7%): 69-year-old men with a Gleason score of eight, a ECOG performance score of two and a SUV_peak_ of the most avid lesion of 9.48. Activity first cycle [^177^Lu]Lu-PSMA-617: 6.2 GBq, activity second cycle [^177^Lu]Lu-PSMA-617: 6.2GBq. TL-PSMA pre-treatment: 260.58 SUV_lbm,peak_*cm^3^, TL-PSMA post-treatment: 2216.94 SUV_lbm,peak_*cm^3^.

**Figure 3 biomedicines-10-01575-f003:**
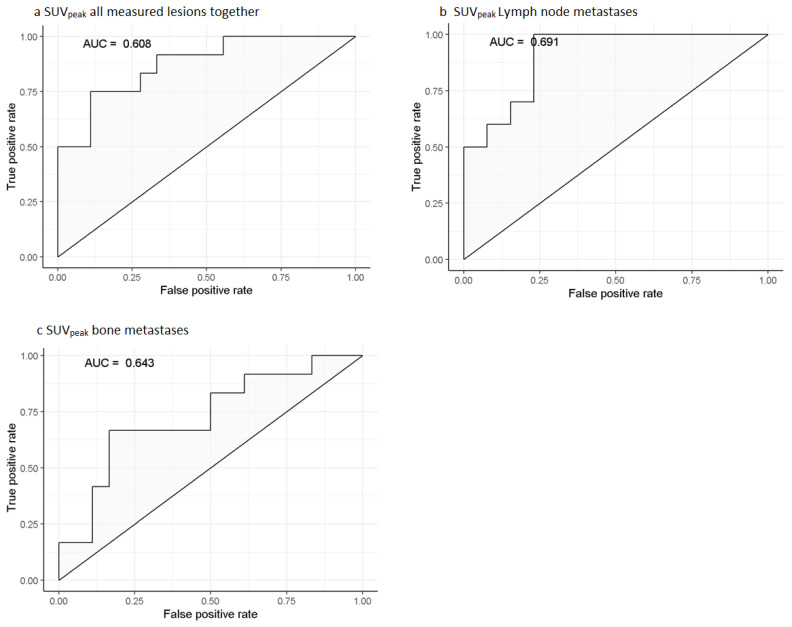
Receiver Operating Characteristics-curves for the predicting of imaging-based response including bootstrap-corrected c-statistic of the three separately tested models in logistic regression. (**a**): SUV_peak_ all measured lesions together, (**b**): SUV_peak_ lymph node metastases, (**c**): SUV_peak_ bone metastases. Legend: AUC = Area under the curve, SUV = Standardized uptake value.

**Figure 4 biomedicines-10-01575-f004:**
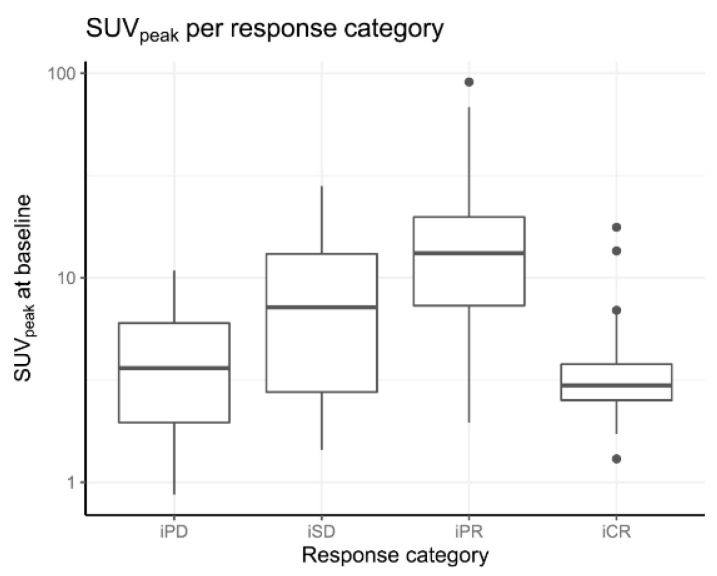
SUV_peak_ values per response category on the tumor-level. Legend: iCR = Imaging complete response, iPD = Imaging progression disease, iPR = Imaging partial response, iSD = Imaging stable disease, SUV = Standardized uptake value.

**Figure 5 biomedicines-10-01575-f005:**
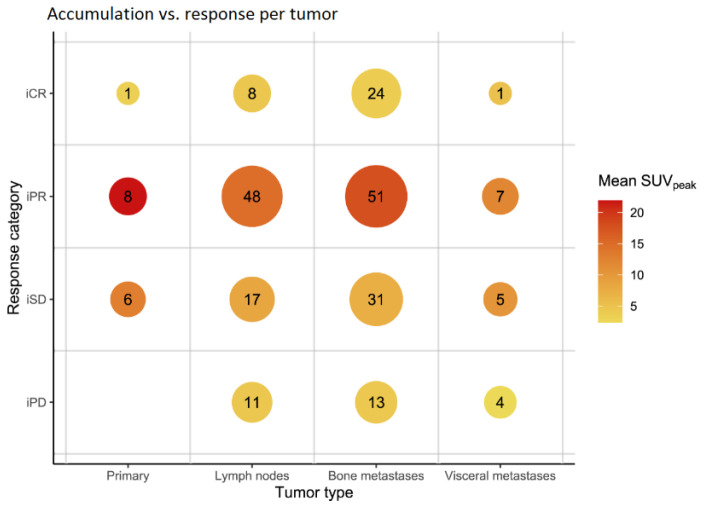
Relationship between metastasis type, response, and accumulation. Numbers in the plot indicate the number of tumors in the corresponding group. Legend: iCR = Imaging complete response, iPD = Imaging progression disease, iPR = Imaging partial response, iSD = Imaging stable disease, SUV = Standardized uptake value.

**Figure 6 biomedicines-10-01575-f006:**
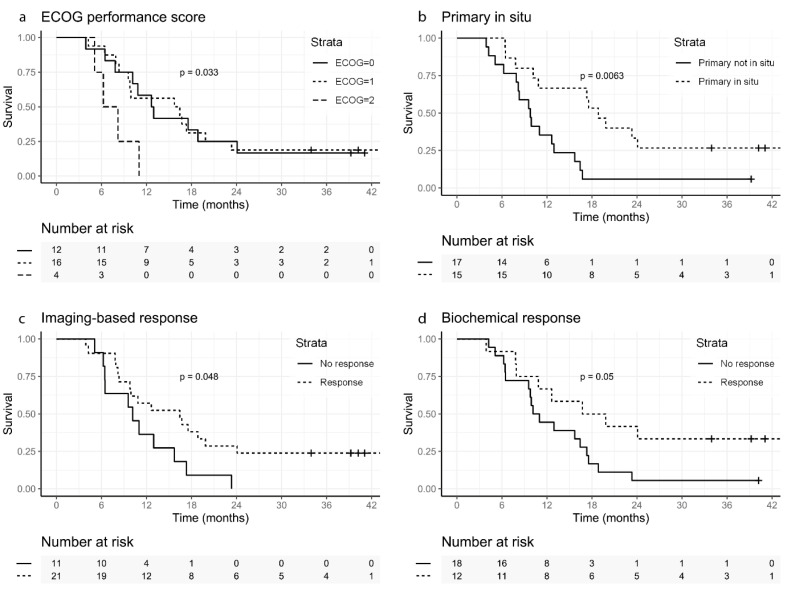
Kaplan–Meier curves showing the significant survival probability, expressed as a percentage, following the first cycle of [^177^Lu]Lu-PSMA-617 treatment. (**a**): ECOG performance score, (**b**): Primary prostate in situ yes or no, (**c**): Imaging-based response (≥30 TL-PSMA reduction) yes or no, (**d**): Biochemical response (>50% PSA reduction) yes or no. Legend: ECOG = Eastern Cooperative Oncology Group.

**Table 1 biomedicines-10-01575-t001:** Baseline patient characteristics.

Characteristic	Value
Patients, number	32
Age, years (mean, SD)	70 (6.75)
Baseline PSA, ng/mL (median, IQR)	210.0 (70.75–547.50) ^a^
Weight, kg (median, IQR)	87 (76.25–95.75)
Gleason-score (following ISUP grade group): number of patients (%)	
- 1	2 (6.2)
- 2/3	4 (12.5)
- 4	5 (15.6)
- 5	14 (43.8)
- Not reported	7 (21.9)
Prior therapy: number of patients (%)	
Surgical resection of primary tumor	15 (46.9)
Docetaxel and/or cabazitaxel	26 (81.3)
Abiraterone and/or enzalutamide	31 (96.9)
^223^Radium	13 (40.6)
ECOG performance score: number of patients (%)	
- 0	12 (37.5)
- 1	16 (50.0)
- 2	4 (12.5)
Regular need for pain medication, number of patients (%)	15 (46.9)
Extension of disease: number of patients (%)	
Lymph node metastasis	24 (75)
Bone metastasis	30 (93.8)
Visceral metastasis	7 (21.9)

^a^ of two patients baseline PSA was missing. PSA was not older than 1 month in the remaining patients. Legend: ECOG = Eastern Cooperative Oncology Group, ISUP = International Society of Urological Pathology, IQR = Inter quartile range, PSA = Prostate specific antigen, SD = Standard deviation.

**Table 2 biomedicines-10-01575-t002:** Imaging baseline parameters.

Characteristic	Baseline	After 2 Cycles of [^177^Lu]Lu-PSMA-617
PSMA-TV, mL (median, IQR)	702.17 (340.54–1376.33)	386.5 (188.8–973.8)
TL-PSMA, SUVlbm,peak*cm^3^ (median, IQR)	3755.54 (1804.3–9435.3)	2112.96 (1102.53–4849.83)
SUVpeak (median, IQR)		
- Primary tumor	13.70 (9.29–20.56)	8.80 (6.13–14.09)
- Lymph node metastases	7.40 (3.94–13.40)	4.54 (2.36–8.85)
- Bone metastases	6.84 (2.92–16.09)	4.07 (1.92–8.09)
- Visceral metastases	7.33 (3.74–13.72)	8.15 (2.77–9.28)
SUVmax (median, IQR)		
- Primary tumor	16.99 (13.03–21.77)	10.69 (7.12–17.20)
- Lymph node metastases	10.54 (6.43–19.61)	6.38 (3.46–11.67)
- Bone metastases	9.92 (4.36–21.05)	5.49 (2.59–10.16)
- Visceral metastases	12.80 (5.34–15.20)	9.75 (3.39–10.79)
Overall survival, months (median, IQR)	10 (7–17)	
Death, number of patients (%)	28 (87.5)	

Legend: IQR = Inter quartile range, PSMA-TV = PSMA tumor volume, PSMA = Prostate specific membrane antigen, SUV = Standardized uptake value, TL-PSMA = Total lesion PSMA.

**Table 3 biomedicines-10-01575-t003:** Imaging-based response on the lesion- and patient-level.

Parameter	iCR (n) %	iPR (n) %	iSD (n) %	iPD (n) %	Total
Patient-level (TL-PSMA)	NA	21 (66%)	6 (19%)	5 (16%)	32
Lesion-level (SUVpeak)					
- Lymph nodes metastases	8 (9%)	49 (57%)	18 (21%)	11 (13%)	86
- Bone metastases	24 (20%)	51 (43%)	31 (26%)	13 (11%)	119
- Visceral metastases	1 (6%)	7 (41%)	5 (29%)	4 (24%)	17
- Primary tumor	1 (7%)	8 (53%)	6 (40%)	0	15
- All lesions ^a^	34 (14%)	115 (49%)	60 (25%)	28 (12%)	237

^a^ Al lesions includes: primary tumor, lymph node, bone and visceral metastases. Legend: iCR = Imaging complete response, iPD = Imaging progression disease, iPR = Imaging partial response, iSD = Imaging stable disease, PSMA = Prostate specific membrane antigen, SUV = Standardized uptake value, TL-PSMA = Total lesion PSMA.

**Table 4 biomedicines-10-01575-t004:** Mixed model of imaging-based response on the lesion-level.

	Coefficient	Exp(coeff)	*p*-Value
**Outcome = log(SUV_peak_)**			<0.001 ^a^
Non-responders	ref		
Responders	0.59 (0.35–0.83)	1.80 (1.42–2.29)	<0.001
**Outcome = log(SUV_max_)**			<0.001 ^a^
Non-responders	ref		
Responders	0.61 (0.38–0.84)	1.84 (1.46–2.31)	<0.001
**Outcome = log(SUV_peak_)**			<0.001 ^a^
iPD	ref		
iSD	0.62 (0.28–0.96)	1.86 (1.32–2.62)	<0.001
iPR	1.33 (1.01–1.66)	3.79 (2.74–5.25)	<0.001
iCR	−0.01 (−0.4–0.38)	0.99 (0.67–1.46)	0.960
**Outcome = log(SUV_max_)**			<0.001 ^a^
iPD	ref		
iSD	0.54 (0.22–0.87)	1.72 (1.24–2.38)	0.001
iPR	1.28 (0.97–1.59)	3.6 (2.64–4.92)	<0.001
iCR	0.01 (−0.37–0.38)	1.01 (0.69–1.47)	0.965

Coefficients of the fixed effects in mixed-model analysis, with a random intercept (SUV_peak_/SUV_max_) per patient. As the outcome data was log-transformed prior to regression, the exponent of the coefficient can be interpreted as the factor of difference in SUV_peak_/SUV_max_ between the corresponding response category and the reference category. ^a^
*p*-values for models calculated by the likelihood ratio test between the model and an empty model. Legend: iCR = Imaging complete response, iPD = Imaging progression disease, iPR = Imaging partial response, iSD = Imaging stable disease, SUV = Standardized uptake value.

**Table 5 biomedicines-10-01575-t005:** Geometric means pre-treatment SUV_peak_ and SUV_max_ of imaging-based response on the lesion-level.

Response Category	SUV_peak_
iPD	3.32 (2.44–4.51)
iSD	6.19 (5.02–7.63)
iPR	12.6 (10.75–14.77)
iCR	3.29 (2.5–4.32)
**Response Category**	**SUV_max_**
iPD	4.99 (3.69–6.74)
iSD	8.58 (6.96–10.57)
iPR	17.96 (15.26–21.15)
iCR	5.03 (3.85–6.57)

Legend: iCR = Imaging complete response, iPD = Imaging progression disease, iPR = Imaging partial response, iSD = Imaging stable disease, SUV = Standardized uptake value.

**Table 6 biomedicines-10-01575-t006:** Univariate logistic regression analyses of response on the patient-level; significant responses in bold.

Parameter	Biochemical Response (PSA Reduction > 50% y/n)	Imaging-Based Response (TL-PSMA Reduction ≥ 30% y/n)
	OR, 95% CI, *p*-value	OR, 95% CI, *p*-value
Baseline PSA (log) (ug/mL)	1.74 (0.859–3.54, *p* = 0.096)	**2.074 (1.043–4.12, *p* = 0.019)**
Age (years)	1.07 (0.951–1.20, *p* = 0.252)	1.06 (0.939–1.196, *p* = 0.336)
Total activity [^177^Lu]Lu-PSMA-617 (GBq)	0.535 (0.234–1.22, *p* = 0.092)	0.966 (0.534–1.748, *p* = 0.910)
ECOG performance score ≥ 1	0.334 (0.069–1.628, *p* = 0.169)	0.577 (0.112–2.979, *p* = 0.506)
Need of pain medication y/n	0.713 (0.144–3.53, *p* = 0.678)	0.315 (0.064–1.557, *p* = 0.151)
Previous [^223^Ra]Ra-dichloride y/n	2.276 (0.412–12.6, *p* = 0.338)	3.597 (0.596–21.7, *p* = 0.138)
Lymph node involvement y/n	0.545 (0.092–3.25, *p* = 0.503)	0.648 (0.102–4.128, *p* = 0.641)
Visceral metastases y/n	0.201 (0.019–2.172, *p* = 0.144)	0.332 (0.054–2.02, *p* = 0.232)
Prostate in situ y/n	1.358 (0.292–6.31, *p* = 0.696)	0.874 (0.183–4.169, *p* = 0.866)
Baseline Hb	0.718 (0.293–1.76, *p* = 0.464)	0.735 (0.334–1.618, *p* = 0.435)
Baseline Plt	0.999 (0.989–1.01, *p* = 0.789)	1.001 (0.991–1.01, *p* = 0.858)
Baseline ALP	0.997 (0.986–1.01, *p* = 0.563)	1.001 (0.992–1.01, *p* = 0.810)
Baseline AST	0.964 (0.888–1.05, *p* = 0.353)	0.973 (0.916–1.034, *p* = 0.369)
Baseline Alb	1.035 (0.677–1.58, *p* = 0.874)	0.865 (0.617–1.213, *p* = 0.381)
Baseline LDH	0.999 (0.994–1.00, *p* = 0.786)	0.998 (0.994–1.003, *p* = 0.494)
Baseline GGT	0.980 (0.947–1.01, *p* = 0.177)	0.984 (0.960–1.009, *p* = 0.158)
SUV_peak_ of primary tumor	1.016 (0.932–1.11, *p* = 0.719)	1.005 (0.916–1.102, *p* = 0.915)
SUV_peak_ of most avid metastasis	1.03 (0.987–1.074, *p* = 0.152)	**1.107 (0.977–1.254, *p* = 0.015)**
SUV_max_ of primary tumor	1.00 (0.933–1.08, *p* = 0.911	0.994 (0.921–1.072, *p* = 0.872)
SUV_max_ of most avid metastasis	1.022 (0.987–1.059, *p* = 0.210)	1.049 (0.985–1.117, *p* = 0.050)
PSMA-TV (L)	0.567 (0.224–1.44, *p* = 0.113)	0.807 (0.501–1.30, *p* = 0.372)
TL-PSMA (SUV_lbm,peak_*m^3^)	1.359 (0.827–2.233, *p* = 0.167)	1.309 (0.823–2.081, *p* = 0.211)
TL-PSMA ≥ 30% y/n	**23,363.18 (0.00–1.34 × 10^28^, *p <* 0.001)**	NA
PSA > 50% y/n	NA	**16,300 (0.00–9.6 × 10^29^, *p* < 0.001**)

No comparison could be made for receiving hormone therapy, and the presence of bone metastases as most of the patients received hormone therapy (31/32) and had bone metastases (30/32). Furthermore, no comparison could be made for ISUP score as the individual groups became too small. All variables were tested in a model which included PSMA-TV as a covariate. *p*-values were calculated using the likelihood ratio test between the model with the variable and the model with only PSMA-TV. Legend: Alb = Albumin, ALP = Alkaline phosphatase, AST = Aspartate aminotransferase, CI = Confidence interval, ECOG = Eastern Cooperative Oncology Group, GGT = Gamma-glutamyltransferase, Hb = Hemoglobin, LDH = Lactic acid dehydrogenase, Lu = Lutetium, NA = Not applicable, Plt = Platelets, PSA = Prostate specific antigen, PSMA = Prostate specific membrane antigen, PSMA-TV = PSMA tumor volume, Ra = Radium, SUV = Standardized uptake value, TL-PSMA = Total lesion PSMA.

**Table 7 biomedicines-10-01575-t007:** Univariate Cox-PH regression on the patient-level; significant *p*-Values in bold.

Parameter	HR	*p*-Value
**PSMA-TV (per liter)**	1.25	0.113
**TL-PSMA (per 1000)**	1.04	0.316
**Highest SUV_peak_**	0.98	0.121
**Highest SUV_max_**	0.99	0.163
**SUV_peak_ ≥ 14.87**	0.42	0.115
**Gleason (following ISUP grade group)**		0.386
1	ref	
2/3	0.27	0.169
4	0.35	0.245
5	0.26	0.090
**Unknown**	0.15	**0.034**
**ECOG performance score**		0.104
0	ref	
1	0.99	0.971
2	4.09	**0.026**
**Lymph node involvement**	0.89	0.800
**Visceral metastases**	1.30	0.565
**Biochemical response (PSA reduction > 50%) = yes**	0.429	**0.047**
**Imaging-based response (TL-PSMA reduction ≥ 30%) = yes**	0.457	0.061

Legend: HR = hazard ratio, PSMA-TV = PSMA tumor volume, PSMA = Prostate specific membrane antigen, SUV = Standardized uptake value, TL-PSMA = Total lesion PSMA.

## Data Availability

The datasets used and/or analyzed during the current study are available from the corresponding author on reasonable request.

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
