# Peer review of "Baseline Imaging Derived Predictive Factors of Response Following [177Lu]Lu-PSMA-617 Therapy in Salvage Metastatic Castration-Resistant Prostate Cancer: A Lesion- and Patient-Based Analysis"

_biomedicines, 2022, doi:10.3390/biomedicines10071575_

Round 1

Reviewer 1 Report

The authors investigated imaging-derived factors of [68Ga]Ga-PSMA-11 PET/CT to predict clinical response, in a lesion- and patient-based analysis, in men with mCRPC 281 receiving two cycles of [177Lu]Lu-PSMA-617. I find this work informative, well presented, and important i in the light of recent approval of [177Lu]Lu-PSMA-617.

The authors should discuss the choice of doses vs. doses used nowadays in the clinical setting, and the influence it has on the overall outcome of the study. 

I would advise authors to share the R code used in modelling in the supplementary materials, given that the ultimate goal of the work is for this predictive model to be implemented in decision making process in the clinic, for instance  to determine if a certain lesion will or will not respond..

In the abstract please write" metastatic castration resistant prostate cancer (mCRPC) patients" instead of  "metastatic prostate cancer patients  (mCRPC)".

Line 67, six should be a write as number just like 7.4.

Line 69, contains space before six. 

Author Response

1. The authors should discuss the choice of doses vs. doses used nowadays in the clinical setting, and the influence it has on the overall outcome of the study.  

Interesting point you mention. After the results of the VISION-trail (an international, open-label, phase 3 trial) 7.4 GBq [177Lu]Lu-PSMA-617 became the most commonly applied or assumed ‘standard’ administered activity.

In our study we treated patients in the pre-VISION era, using either an activity of 6.0 GBq or 7.4 GBq. Scientific evidence of which activity is the best choice to use is lacking. There is one study from Rathke et al. (https://doi.org/10.2967/jnumed.117.194209) conducted in 2018 comparing 4,6, 7.4 and 9.3 GBq of [177Lu]Lu-PSMA-617 in terms of administered activity and response relationship. They found no major differences among the 4 different activities and PSA response, 8 weeks after the first treatment. However, they also conclude that because of the limited numbers of patients, it is statistically not reasonable to draw a final conclusion.

These findings are in line with non-published data from our centre (comparing outcomes after 6.0 vs 7.4 GBq/cycle), revealing no significant differences in adverse events, imaging-based response and biochemical response.

Due to the lack of evidence, we have chosen not to address this in the discussion.

2. I would advise authors to share the R code used in modelling in the supplementary materials, given that the ultimate goal of the work is for this predictive model to be implemented in decision making process in the clinic, for instance  to determine if a certain lesion will or will not respond.

Thank you for your advice. We will share the R code used in modelling in the supplementary materials. We will refer to that in the statistical analyses section as appendix B.

3. In the abstract please write" metastatic castration resistant prostate cancer (mCRPC) patients" instead of  "metastatic prostate cancer patients  (mCRPC)".

Thank you for your comment, it has been changed to metastatic castration resistant prostate cancer.

4. Line 67, six should be a write as number just like 7.4.

We have changed this.

5. Line 69, contains space before six. 

The extra space has been removed.

Reviewer 2 Report

In this manuscript the authors evaluated  whether imaging-derived factors on baseline PSMA PET/CT can predict response on Lu-PSMA-617, on a lesion- and patient-level analysis, in men with mCRPC.

Objective response after two cycles of Lu-PSMA-617 on lesion- and patient-level was based on PERCIST criteria. 32 patients were included in the final anaylsis.

Overall intersting results in this field of PCa research. Methods and statistic behind seem solid and the study brings a clear clinical message. Limitations are clearly stated in the appropriate paragraph.

However, I have one more comment before acceptance.

The authors would underline more the risk stratification extrapolated by these findings in the dedicated paragraph. In case of low (pre) accumulation what are the available strategies?

Author Response

1. The authors would underline more the risk stratification extrapolated by these findings in the dedicated paragraph. In case of low (pre) accumulation what are the available strategies?

Thank you for your comment to underline more the risk stratification. We choose not to incorporate any suggestions on risk stratification, because all the included patients in this cohort have end stage disease (failing regular treatment lines). Of course based on our results, we would advise not give  [177Lu]Lu-PSMA-617 therapy if insufficient [68Ga]Ga-PSMA-11 accumulation is seen on the [68Ga]Ga-PSMA-11 PET/CT. However as discussed in the discussion our study results are preliminary and based on a small sample size. We wish to validate our findings in a larger cohort and future prospective studies. At the moment the data is too immature and limits the ability to draw definite conclusions.